# Tunable Nonlinear Optical Property of MnS Nanoparticles with Different Size and Crystal Form

**DOI:** 10.3390/nano10010034

**Published:** 2019-12-21

**Authors:** Zhihao Zhang, Pengchao Li, Yuzong Gu

**Affiliations:** Institute of Micro/Nano Photonic Materials and Applications, School of Physics and Electronics, Henan University, Kaifeng 475004, China; kfzzh@163.com (Z.Z.); pengchaoli158566@163.com (P.L.)

**Keywords:** MnS, nanoparticles, Z-scan, nonlinear enhanced

## Abstract

It is significant to study the reason that semiconductor material has adjustable third-order optical nonlinearity through crystal form and dimensions are changed. αMnS nanoparticles with different crystal forms and sizes were successfully prepared by one-step hydrothermal synthesis method and their size-limited third-order nonlinear optical property was tested by Z-scan technique with 30 ps laser pulses at 532 nm wavelength. Nanoparticles of different crystal forms exhibited different NLO (nonlinear optical) responses. γMnS had stronger NLO response than αMnS because of higher fluorescence quantum yield. Two-photon absorption and the nonlinear refraction are enhanced as size of nanoparticlesreduced. The nanoparticles had maximum NLO susceptibility which was 3.09 × 10^−12^ esu. Susceptibility of αMnS increased about nine times than that of largest nanoparticles. However, it was reduced when size was further decreased. This trend was explained by the effects of light induced dipole moments. And defects in αMnS nanoparticles also had effect on this nonlinear process. MnS nanoparticles had potential application value in optical limiting and optical modulation.

## 1. Introduction

With the development of the information age, people have increasing requirements for information transmission, storage, exchange, or display speed in communication network systems [1]. However, a large number of electronic devices, electro-optical, and optical-electrical conversion devices in systerm limit the speed of information transmission [2,3]. Improving the speed of information transmission becomes the most urgent need. It is the only way to break this bottleneck that faster response photonic devices replace electro-optical and photo-electrical conversion devices [4,5,6]. In recent years, semiconductor nanomaterials are one of the rapidly developing research fields because of their potential applications in optoelectronic devices, all-optical switches, and fluorescent markers. Lots of study reported that pure CdSe quantum dots and ZnSe quantum dots [7,8], semiconductor-particles-doped CdSe/CdS quantum dots [9], and Mn-doped PbSe quantum dots [10] all show excellent NLO (nonlinear optical) response.

CdS, CuS, ZnS, MnS and etc. are typical semiconductor nanomaterials and exhibit good electrochemical property [11,12,13,14]. Meanwhile, these typical sulfide nanoparticles also exhibit significant two-photon absorption and refraction in terms of NLO property [15,16,17,18]. Enhanced NLO response of these semiconductor nanomaterials was due to intense surface plasmon resonance [7]. Cadmium, zinc, copper, and manganese are significant metals ions for preparation of photoelectrons and NLO nanoparticles, which may apply in optoelectronic devices and non-linear devices [7,8,16,18]. Some studies have experimentally determined that quantum dots exhibited the relationship between NLO enhancement and quantum dot size [19,20,21,22]. Although there were many explanations for excellent NLO performance of quantum dots and mechanism for their performance enhancement, quantum dot synthesis process was complicated and the samples’ stability was weaker compared with nanoparticles, resulting in their limitation for preparation of faster response photonic devices. In addition, the relationship between nanoparticles’ size and nonlinear optical property was still lack. The correlation between size of semiconductor nanoparticles and their three order susceptibility has not been well studied. The mechanism for studying NLO property of nanoparticles is urgent and significant for the application of NLO devices.

In the previous study, we synthesized different crystal form MnS nanoparticles and their graphene composites [18,23]. We discussed the relationship between synthesis time of composites and NLO property, and the relationship between graphene addition of composites and NLO property. However, the relationship between crystal form of pure MnS nanoparticles and NLO response, and the relationship between size and NLO response have not been studied. The mechanism of NLO susceptibility changed is still unclear. So it is necessary to explore the relationship between size of nanoparticles and NLO property. Therefore, in this study, we used one-step hydrothermal method to synthesize MnS nanoparticles with different sizes and γMnS nanoparticles. The goal is to obtain MnS nanoparticles of tunable NLO property and achieve products which can be applicated in optical limiting and optical modulation by clarifying the mechanism of NLO susceptibility.

## 2. Experiments

### 2.1. Synthesis of αMnS

MnS was prepared by the facile one-step hydrothermal method [18,23]. This method is easy to operate, has high production efficiency, and can produce specific crystal forms of MnS or other semiconductor materials by changing temperature, synthesis time, kinds of solvent, and sulfur source. The method is important for the potential application of semiconductor nanomaterials in the fabrication of NLO devices.

An appropriate amount of ethylene glycol was used as solvent containing manganese source and sulfur source. Thioacetamide (TAA) was used as sulfur source and MnCl_2_·4H_2_O as manganese source. First, TAA and MnCl_2_·4H_2_O were thoroughly mixed and added to ethylene glycol. Next, the solution was stirred through magnetic stirrer and sonicated for 1 h. Then, the solution was placed in Teflon liner. The Teflon liner was transferred to stainless steel autoclave and reacted at 190 °C for 6 h. Finally, the product was washed several times with absolute ethanol and deionized water. The final product was dried in vacuum oven at 45 °C for 48 h and labeled as αMnS-6. Five samples were prepared by the same steps with different reaction times. The synthesis times were 6 h, 8 h, 10 h, 12 h and 14 h, respectively. This was basically consistent with the synthetic methods in our previous studies. Five samples were labeled as αMnS-6, αMnS-8, αMnS-10, αMnS-12 and αMnS-14, corresponding to synthesis times of 6 h, 8 h, 10 h, 12 h and 14 h. However, this method synthesized αMnS bulk crystals instead of nanoparticles. And nanoparticles are condensed not scattered. Nanoparticles make up bulk crystals, so the massive crystals need to be dissolved in the solution with sufficient ultrasound to obtain nanoparticles. αMnS powder was first dissolved in ultrapure water. The solution was then thoroughly stirred and sonicated for 4 h and product was dried in a vacuum oven at 40 °C for 24 h. At this time, massive αMnS became αMnS nanocrystal. γMnS nanoparticles could be obtained by same steps, synthesis time was 6 h with different heat temperature, which was 170 °C.

### 2.2. Instrumental Characterization

XRD patterns of αMnS and γMnS were obtained on X-ray diffraction (XRD, Bruker D8 Advance, Bruker Inc., Karlsruhe, badensko-wuertembersko, Germany). SEM images of samples were acquired by scanning electron microscope (SEM, Carl Zeiss Inc., Oberkochen, Baden-Württemberg, Germany). αMnS and γMnS was also tested by Bruker Optics Vertex 70 (Bruker Inc., Karlsruhe, badensko-wuertembersko, Germany) and Ultraviolet–Visible absorption instrument (Uv-Vis, Cary 5000, Agilent Inc., Sacramento, CA, USA), respectively. Raman spectra were obtained on Renishaw inVia (Renishaw Inc., Gloucester, Gloucestershire, UK). The Z-scan patterns were received on picosecond laser (picosecond laser, PLA2251A, Ekspla Inc., Vilnius, Lithuania) with wavelength 532 nm and pulse width 30 ps. The incident laser wavelength and pulse width used to obtain the z-scan curves are consistent with our previous studies.

## 3. Results and Discussion

### 3.1. Structure and Morphology Characterization

Samples were exposed to X-ray diffraction to obtain XRD spectra. XRD patterns of αMnS-6, αMnS-10 and αMnS-14 were shown in Figure 1. For pure αMnS nanoparticles, there was no diffraction peaks of impurities detected, demonstrating that products were pure αMnS. The characteristic diffraction peaks of αMnS are located at 29.6°, 34.3°, 49.3°, 59.3°, 58.5°, and 61.4°, corresponding to (111), (200), (220), (311), and (222) [24]. The sharpness of diffraction peak in XRD spectra represented the crystallinity of samples [25]. higher rystallinity of αMnS was, sharper diffraction peak was. Figure 1 showed that sharpness of the diffraction peak of αMnS-14 was higher than that of αMnS-6 and αMnS-10, indicating that the crystallinity of αMnS-14 was the highest. Due to different crystal form, the peaks of γMnS were located at 26°, 28°, 29°, 46°, 50°, and 53°, corresponding to (100), (002), (101), (110), (103), and (112) [23]. For γMnS, sharpness of diffraction peak was lower than αMnS, demonstrating that γMnS exhibited lower crystallinity.

As mentioned above, MnS crystal synthesized by the hydrothermal method with sufficient agitation and ultrasonication to obtain nanoparticles. Figure 2a showed the aggregation of αMnS crystals. Figure 2b,c shows that MnS crystal was bulky, and volume of crystal block became large as synthesis time increased. αMnS nanoparticles obtained after treatment [18]. The volume of nanoparticles decreased as the synthesis time increased and size range was 40 nm to 200 nm [23]. This might be due to high temperature and high pressure in stainless-steel reactor. Initially, manganese source and sulfur source combined to form manganese sulfide nanocrystals, which were then continuously formed massive αMnS. Firstly,αMnS formed was loose cluster of grapes. Then, as synthesis time increased, αMnS of massive structure began to become dense, presumably because of environmental influence forming massive crystal, resulting in volume of αMnS nanoparticles reduced. Along with the continuous formation of αMnS nanoparticles, volume of the crystals was increased, but volume of nanoparticles continuously was reduced under high temperature and high pressure. Figure 2d showed mapping images of MnS. Green areas represented sulfur and red areas represented manganese, which indicated that the product was MnS not another substance. Figure 2e displayed the relationship between nanoparticles and bulk crystals under high temperature and high pressure environment as the synthesis time extended.

It could be seen from Figure 3 that αMnS-6 showed strong exciton absorption peak at 285 nm, αMnS-10 showed exciton absorption peak at 278 nm, the absorption peak of αMnS-14 was at 272 nm, and the absorption peak of γMnS was at 281 nm. As reaction time increased, absorption spectra exhibited significant blue shift, indicating that diameter of nanoparticles was decreased, which was consistent with the results as mentioned above. The blue shift of 10 nm indicated some changes in electronic state of nanoparticles. These changes in electronic states might be related to NLO property of αMnS.

Figure 4 showed Raman spectra of MnS with different crystal forms. *α*MnS exhibited Raman absorption peak at 636 cm^−1^. However, Raman absorption peak γMnS was at 646 cm^−1^. There was a little difference between peak of *α*MnS and that of γMnS. The reason for difference might be that Raman absorption peak shift should be affected by lattice constant of distortion.

### 3.2. NLO Property of Nanoparticles

In our previous study, NLO response of *α*MnS/rGO and γMnS/rGO was investigated. And results showed that nonlinearity of graphene composites was enhanced and the reason for NLO enhancement was surface defects and synergistic effect, including local field theory and charge transfer [18,23]. In this study, NLO absorption and refraction of *α*MnS with different size and γMnS were measured by Z-scan technique using single Gaussian beam. The Nd:YAG laser system used for excitation generated 30 pisecond laser pulse at 532 nm wavelength. Laser system generated repetition frequency of 10 Hz and had beam waist radius of about 10.6 μm at the focal plane. CS_2_ was standard for Z-scan curves, which could calibrate curves, so that the center of the curve is centered on *Z*-axis. The measurement data could ignore the absorption and scattering effects of the sample. The samples on a moving platform move along the *Z*-axis and focal plane of 250 mm focal length lens. With absolute ethanol as the solvent, the sample concentration was 0.125 mg/mL.

Figure 5a,b showed typical OA (open aperture) and CA (close aperture) /OA Z-scan curves for nanoparticles of different crystal forms. The OA curve showed valley indicating two-photon absorption process and the positive nonlinear absorption coefficient β. The CA/OA Z-scan curves displayed peak-to-valley profile, indicating self-focusing and positive nonlinear refractive index n_2_ of nanoparticles. The Reχ^(3)^ and Imχ^(3)^ values could be obtained by the equations showed below.

When incident light was incident on the sample, normalized transmittance of Z-scan in actual measurement could be represented by [25,26]
(1)$$T(z)\;=\overset{\infty }{\underset{m\;=\;0}{\sum }}\left\{{\left[q_{0}\left(z\right)\right]}^{m}/{\left(1\;+\;m\right)}^{3/2}\right\}$$
where q_0_(z) was calculated according to q_0_(z) = βI_0_Leff/(1 + z^2^/z_0_^2^) [26]. β could be obtained by
(2)$$\beta \;=\;\left[2\sqrt{2}\left({1\;-\;T}_{z\;=\;0}\right)\left({1\;+\;Z}^{2}{\;+\;Z}_{0}^{2}\right)\right]{/(I}_{0}L_{\mathrm{eff}}).$$

L_eff_ could be acquired by the formula below which was effect length.
L_eff_ = (1 − exp( −αL))/(αL)(3)

Imaginary part and real part could be calculated by the formulas that Imχ^(3)^ = cn_0_λβ/480π and Reχ^(3)^ = n_0_n_2_/3π, where n_2_ was nonlinear refractive index obtained by
n_2_ = (2.941 × 10^6^λω_0_n_0_τ△T_p-v_)/[EL_eff_(1 − S)^0.25^](4)

So nonlinear susceptibility of samples could be calculated by [27]
(5)|χ(3)| = [(Reχ(3))2 + (Imχ(3))2]1/2

The third-order susceptibility of nanoparticles was also calculated in our previous studies, but there was no discussion on various sizes and mechanisms of NLO property of nanoparticles [18,23]. According to equations, increase of α and L_eff_ resulted in NLO absorption coefficient β enhanced. ΔT_p-v_ would influence change of n_2_ and ultimately affected final trend NLO susceptibility χ^(3)^. T could be observed from Figure 5a,b that T = 0.86 and T = 0.83. And ∆T_p-v_ also could be obtained by Figure 5a,b that ∆T_p-v,αMnS-6_ = 0.29 and ∆T_p-v,__γ__MnS_ = 0.3. β could be calculated by the equations that β_αMnS-6_ = 1.78 × 10^−11^mW^−1^ and β_γ__MnS_ = 2.14 × 10^−11^mW^−1^. And n_2_ could be obtained by equations that n_2,αMnS-6_ = 2.17 × 10^−12^ esu and n_2,__γ__MnS_ = 2.48 × 10^−12^ esu. Reχ^(3)^, Imχ^(3)^, and χ^(3)^ are displayed in Figure 6. Susceptibility χ^(3)^ of γMnS was 0.95 × 10^−12^ esu and χ^(3)^ of αMnS was 0.33 × 10^−12^ esu. Due to different crystal forms, αMnS and γMnS nanoparticles exhibited different NLO response.

It was known from our previous studies that fluorescence peak of γMnS appeared at 432 nm and the excitation wavelength was 283 nm [23]. However, αMnS did not show fluorescence peak [18]. The decrease in fluorescence quantum yield might weaken the NLO characteristics of the sample. αMnS might have more non-radiative defects than γMnS due to the dissipation of energy by the lattice thermal vibration, which caused the surface-localized electrons to quench rapidly due to non-radiative defects, greatly reducing the fluorescence of semiconductor structures quantum yield, which resulted in weak NLO performance.

It could be obtained from our previous study that αMnS with different synthesis time exhibited two-photon absorption [18]. Value of T could be acquired from our previous study [18] that T = 0.83, T = 0.71, T = 0.65, T = 0.69, and T = 0.84. ∆T_p-v_ could be obtained from our previous study that ∆T_p-v,αMnS-6_ = 0.29, ∆T_p-v,αMnS-8_ = 0.42, ∆T_p-v,αMnS-10_ = 0.5, ∆T_p-v,αMnS-12_ = 0.81, and ∆T_p-v,αMnS-14_ = 0.53 [18]. β could be calculated by the equations that β_αMnS-6_ = 1.78 × 10^−11^mW^−1^, β_αMnS-8_ = 2.17 × 10^−11^mW^−1^, β_αMnS-10_ = 3.73 × 10^−11^mW^−1^, β_αMnS-12_ = 4.52 × 10^−11^ mW^−1^, and β_αMnS-14_ = 3.98 × 10^−11^ mW^−1^. n_2_ could be obtained by equations that n_2,αMnS-6_ = 2.17 × 10^−12^ esu, n_2,αMnS-8_ = 3.18 × 10^−12^ esu, n_2,αMnS-10_ = 3.84 × 10^−12^ esu, n_2,αMnS-12_ = 6.31 × 10^−12^ esu. And n_2,αMnS-14_ = 4.08 × 10^−12^ esu. Reχ^(3)^, Imχ^(3)^, and χ^(3)^ are displayed in Figure 6.The nonlinear absorption characteristics exhibited by the nanoparticles first increased and then decreased with size of the nanoparticles decreased. And Figure 6 showed all samples of αMnS exhibited positive nonlinear refractive index. The nonlinear optical parameters of all samples in Figure 6 were based on the above formula. NLO response of αMnS nanoparticles increased as their size decreased. The nonlinear refraction characteristics of αMnS also decreased with the increase of size, which showed that changing size by controlling synthesis time could control nonlinear characteristics of nanoparticles. Meanwhile, it could be seen from Figure 6 that changing the temperature to control crystal form of nanoparticles could also change nonlinear characteristics of nanoparticles. The nonlinear response of nanoparticles became controllable, which made them have application potential in the fabrication of nonlinear devices. It could be seen from Figure 6 that susceptibility of αMnS decreased instead when nanoparticles were further reduced. The best result for χ^(3)^ of αMnS-12 was 3.09 × 10^−12^ esu and αMnS-6 was 0.33 × 10^−12^ esu, demonstrating that χ^(3)^ of αMnS-12 increased about nine times larger than that of αMnS-6. αMnS nanoparticles with different synthesis times showed tunable NLO response.

The photoinduced transition dipole moment determined the NLO performance in αMnS nanoparticles. The intensity of the exciton oscillation was related to overlap of electron and hole wave functions [27]. Since movement of electrons and holes was free, the electron and hole wave functions of two excitons hardly overlap when size of nanoparticles was large and thus the photoinduced dipole moment was small. As the size of nanoparticles decreased, electrons might be limited by size of the nanoparticles, and the wave functions of electrons and holes in the two pairs of excitons became more overlapped as size decreased. This enhanced oscillation intensity of two excitons and improved photoinduced dipole moment and third-order NLO property of nanoparticles [28,29]. However, as size of nanoparticles was further reduced, the holes and electrons in two excitons were limited, resulting in nonlinear characteristics of nanoparticles decreased.

On the other hand, defects in MnS nanoparticles possibly reduced their optical nonlinearity [30]. The large ground-state dipole moment was an intrinsic internal field, which was related to the defects of nanoparticles and would strongly influence selection rules and electronic structures through exciton transitions [30]. Due to reduced spatial overlap of local electron and hole wave functions, radiation defects could enhance the photodipole moment. Surface-localized electrons were rapidly quenched for non-radiative defects, which greatly reduced fluorescence quantum yield of semiconductor structure. NLO performance was related to defects in nanoparticles and non-radiative defect state in nanoparticles was negative for nonlinearity, which could quickly quench electrons and reduce the spatial overlap of electron and hole wave functions. This reduced light induced dipole moment, resulting in NLO response of nanoparticles weakened.

## 4. Conclusions

In summary, αMnS and γMnS nanoparticles were successfully prepared by simple one-step hydrothermal synthesis method. Structure and morphology of nanoparticles were acquired on XRD, SEM, UV absorption spectra, and Raman spectra, and their tunable and size-limited three order nonlinear optical property was tested by Z-scan technique. By controlling synthesis temperature and time to change crystal form and size of nanoparticles, nanoparticles of different sizes had different nonlinear characteristics. Higher fluorescence quantum yield resulted in stronger NLO response for γMnS. As size was reduced, NLO response of αMnS first increased and then reduced. αMnS nanoparticles with 12 h synthesis time have maximum NLO susceptibility. This could be explained by the effects of light induced dipole moments and defects of nanoparticles. Tunable nonlinearity makes MnS potentially useful in NLO device manufacturing.

## Figures and Tables

**Figure 1 nanomaterials-10-00034-f001:**
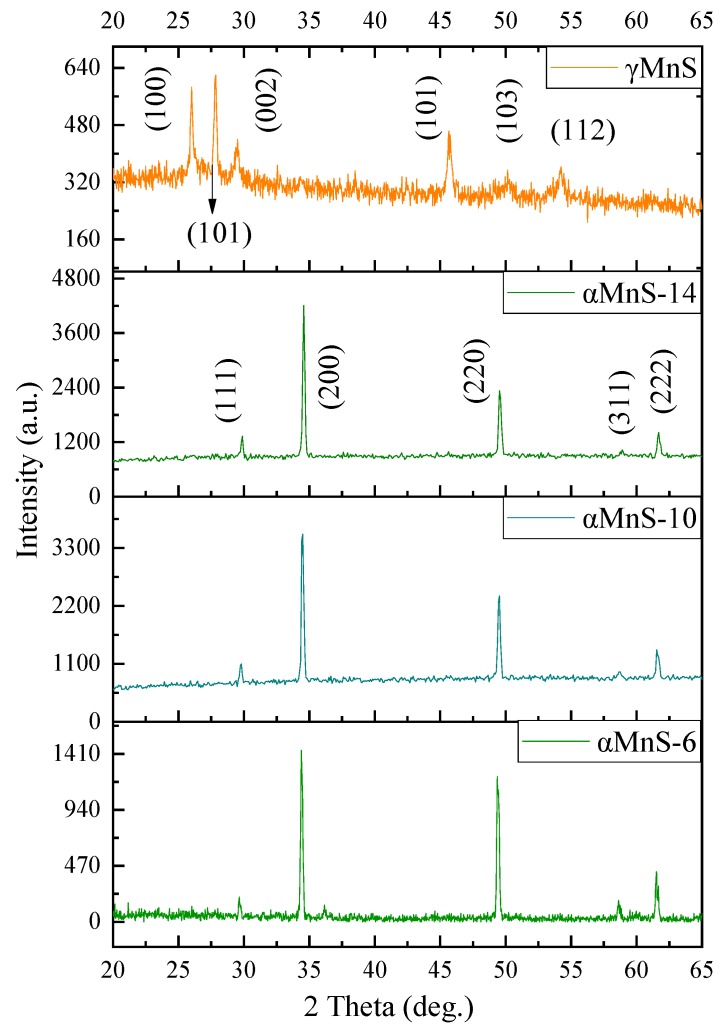
XRD patterns of αMnS-6, αMnS-10, αMnS-14 and γMnS.

**Figure 2 nanomaterials-10-00034-f002:**
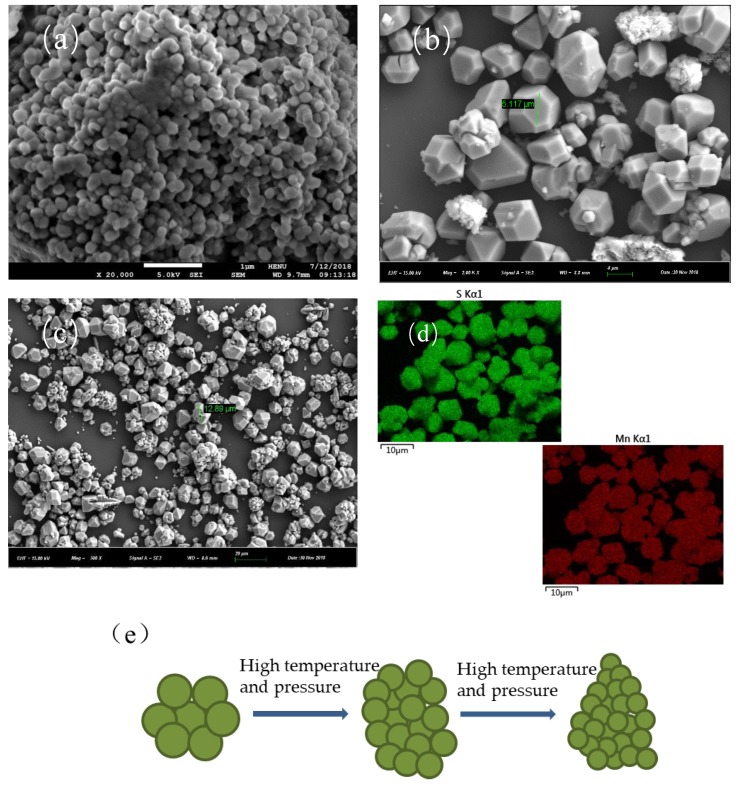
SEM image of (**a**) massive αMnS-6, (**b**) massive αMnS-10, (**c**) massive *α*MnS-14, (**d**) mapping images of MnS, and (**e**) process of αMnS bulk crystals change.

**Figure 3 nanomaterials-10-00034-f003:**
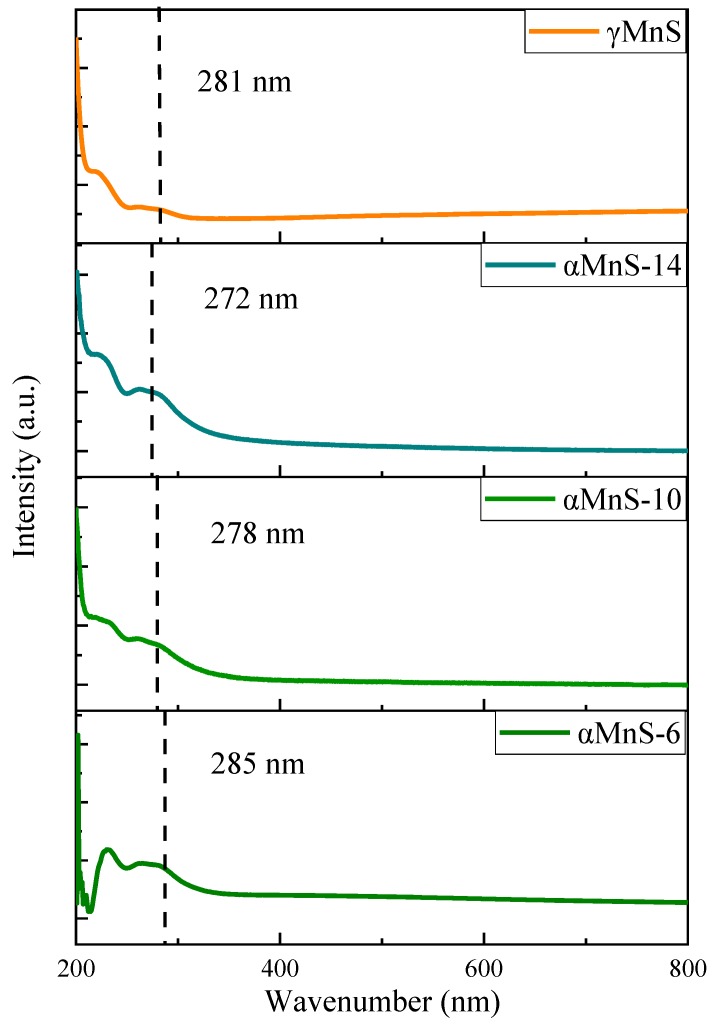
UV absorption spectra of αMnS-6, αMnS-10, and *α*MnS-14.

**Figure 4 nanomaterials-10-00034-f004:**
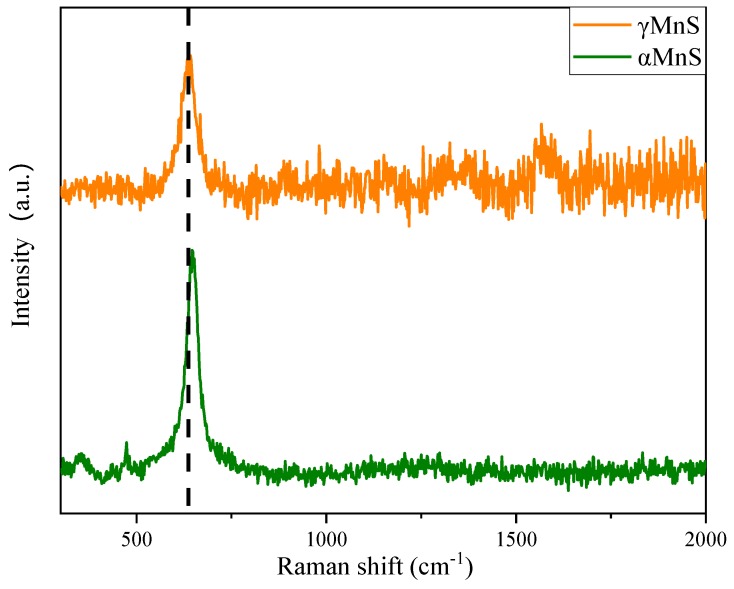
Raman spectra of *α*MnS and γMnS.

**Figure 5 nanomaterials-10-00034-f005:**
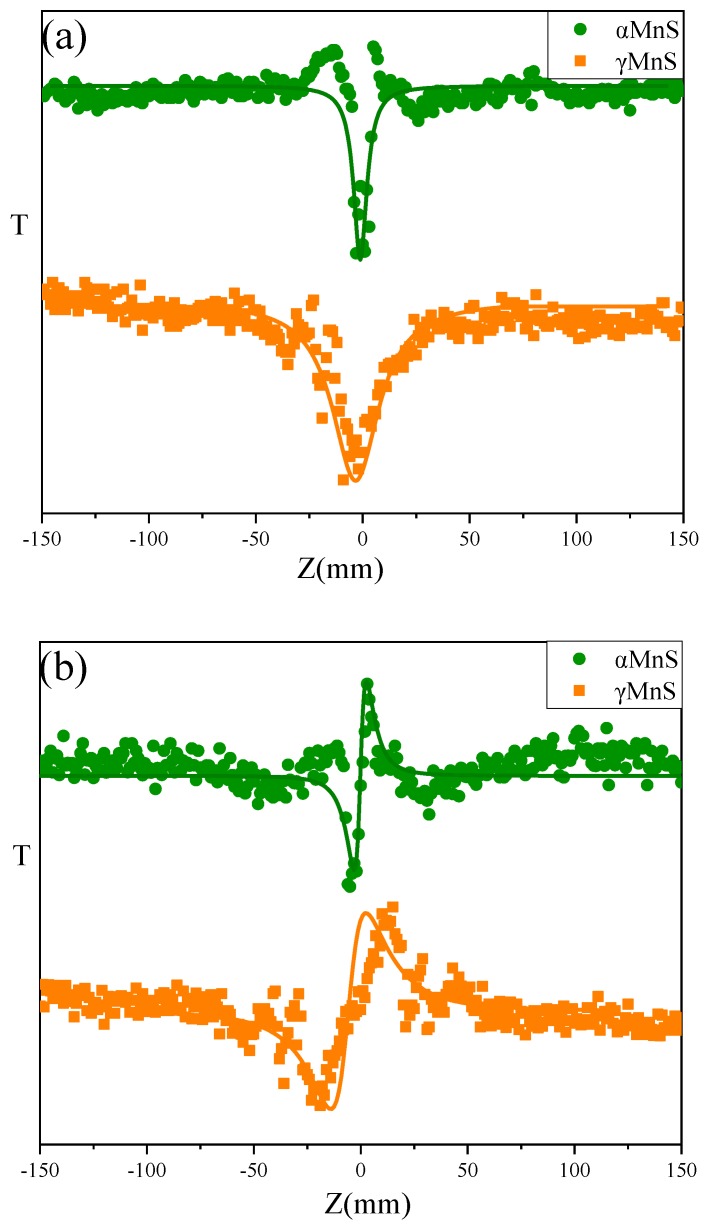
(**a**) OA Z-scan curves of *α*MnS and γMnS. (**b**) CA/OA Z-scan curves of *α*MnS and γMnS.

**Figure 6 nanomaterials-10-00034-f006:**
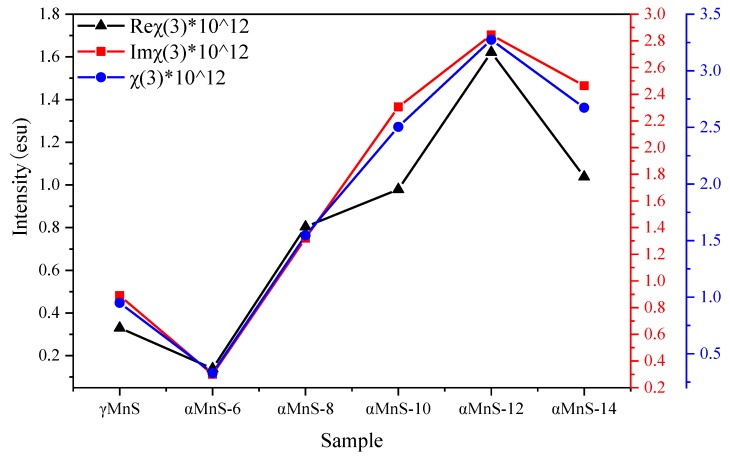
The nonlinear optical parameters of all samples.

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
