# Peer review of "Tunable Nonlinear Optical Property of MnS Nanoparticles with Different Size and Crystal Form"

_nanomaterials, 2019, doi:10.3390/nano10010034_

Round 1
Reviewer 1 Report
The authors have investigated the relation ship between size of MnS particales and the nonlinear optical responses. The results indicates
useful information for utilizing the NLO of MnS nanoparticlses. On the othre hand I have some concernes as follows.
1. Why the authors focused on both alpha and gamma MnS particles? What is the expected qualitative difference between two crystalline types? Moreover, what happens when the sizes of gamma MnS particles are controlled as the case of alpha MnS?
2. Did the authors estimated the sizes of alpha and gamma nano particles? I’m not sure there seizes are sufficiently small to make appropriate discussion about the overlapping of electron and hole wave functions.
3. Line 77
Does "synsesized a-MnS bulk crystal instead of nanoparticles" mean "nanopariticles are condensed"? The meaning is not unclear, so detailed explanation is beneficial.
4. Line 101
The authors say that “For pure a-MnS nano particles, there was no diffraction peak detected.” I don’t know what this means because Fig. 1 shows the diffraction peaks of aMnS-6, 10, and14 nano particles.
5. Line 118
The authors claim that “… the volume of the nanoparticles decreased as the synthesis time increased….” However, when I see Fig. 2(a)-(c), I find that the size becomes bigger for increasing the time 6h, 10h, and 14hr. Lines 124-126 might explain this point, but it’s difficult to understand the situation. Some schematic illustration or some example SEM images might be necessary for understanding the MnS particle preparation.
6. Line 143
The authors mention that the reason of the difference of Raman shift is due to “various structure and electronic states.” I think Rama shift should be affected by lattice constant of distortion.
7. In Fig. 5, what is the physical meaning of broadening of dips (or peak-valley transition) in gamma-MnS compared with alpha-MnS?
8. Line 186
The author says that the reason for weaker NLO performance of a-MnS is due to the different crystal form. But this is equivalent to not saying anything. Are there more meaningful discussion? I’m not sure but are there some influence of defects and so on?
9. Line 187
The authors say that “gamma-MnS was not as stable as that of alpha-MnS”. But I understood that the NLO of gamma MnS is better than alpha MnS. What does this sentence mean, or what ‘stable’ mean?
10. In Fig. 6(a), for the a-MnS 10 and12, there seems many dips around Z=0. Does this has physical meaning or show merely noise?
11. In Fig. 7, how the result of gamma-MnS compared with alpha MnS is expressed?
12. Minor points
Line 126 “Figure 2(g) and (h)” -> “Figure 2 (d)”
Line 132 “a-MnS-10 was 272 nm” -> “a-MnS-14 was 272 nm”
Line 158 “OA Z-scan OA Z-scan” -> “OA Z-scan”
Author Response
Dear reviewer,
Thank you for your letter and for the comments concerning our manuscript entitled “Tunable nonlinear optical property of MnS nanoparticles with different size and crystal form” (ID: nanomaterials-666187). Those comments are all valuable and very helpful for revising and improving our paper, as well as the important guiding significance to our researches. We have carefully studied the comments and have made correction which we hope to meet with approval. Revised portions are marked in red in the manuscript. The main corrections in the manuscript and the responds to the reviewer’s comments are as follow.
Why the authors focused on both alpha and gamma MnS particles? What is the expected qualitative difference between two crystalline types? Moreover, what happens when the sizes of gamma MnS particles are controlled as the case of alpha MnS?
Response 1: When we synthesized α-MnS, we found that crystal form of product changed when temperature was changed. Therefore, comparison of different crystal forms of MnS was performed in the manuscript. α-MnS is more structurally stable than gamma MnS. In fact, γ-MnS has very good fluorescence characteristics, but fluorescence characteristics weaken with different temperature. When the synthesis temperature is 170 ° C, the size of gamma MnS nanoparticles may also change if the time is prolonged, but too long synthesis time may make γ-MnS nanoparticles translate to α-MnS which is more stable.
Did the authors estimated the sizes of alpha and gamma nano particles? I’m not sure there seizes are sufficiently small to make appropriate discussion about the overlapping of electron and hole wave functions.
Response 2: The size of nanoparticles is within 200nm, the smallest size of nanoparticles is 40nm, and the overlap of the electron and hole wave functions can explain nonlinear change of nanoparticles. We maintain that the overlap of electron and hole wave functions can indeed explain non-linear change trend. At the same time, the defects of nanoparticles also have significant impact on nonlinearity
Line 77 Does "synsesized a-MnS bulk crystal instead of nanoparticles" mean "nanopariticles are condensed"? The meaning is not unclear, so detailed explanation is beneficial.
Response 3: We have added in the manuscript.
Nanopariticles are condensed and not scattered. Nanoparticles make up bulk crystals, so the massive crystals need to be dissolved in the solution with sufficient ultrasound to obtain nanoparticles.
Line 101 The authors say that “For pure a-MnS nano particles, there was no diffraction peak detected.” I don’t know what this means because Fig. 1 shows the diffraction peaks of aMnS-6, 10, and14 nanoparticles.
Response 4: This is our mistake. We mean that there was no diffraction peaks of impurities detected. And we make change in the manuscript.
Line 118 The authors claim that “… the volume of the nanoparticles decreased as the synthesis time increased….” However, when I see Fig. 2(a)-(c), I find that the size becomes bigger for increasing the time 6h, 10h, and 14hr. Lines 124-126 might explain this point, but it’s difficult to understand the situation. Some schematic illustration or some example SEM images might be necessary for understanding the MnS particle preparation.
Response 5: The SEM images in the manuscript are bulk crystals. The volume of bulk crystals increases with synthesis time extended. The three images below are SEM images of the nanoparticles, which have been used in previously published articles. But we have not explored the reason why the diameter of the nanoparticles becomes small, so we will elaborate in this manuscript.
We added flow chart as follow to describe changes in size ofnanoparticles.
Line 143 The authors mention that the reason of the difference of Raman shift is due to “various structure and electronic states.” I think Rama shift should be affected by lattice constant of distortion.
Response 6: We corrected to "The reason for difference might be that Rama absorption peak shift should be affected by lattice constant of distortion."
In Fig. 5, what is the physical meaning of broadening of dips (or peak-valley transition) in gamma-MnS compared with alpha-MnS?
Response 6: Broadening of dips and peak-valley transition are related to PS lasers. Nonlinearity of material has less effect on them.
Line 186 The author says that the reason for weaker NLO performance of a-MnS is due to the different crystal form. But this is equivalent to not saying anything. Are there more meaningful discussion? I’m not sure but are there some influence of defects and so on?
Response 8: We remove that sentence and add the content in manuscript as follow.
αMnS might have more non-radiative defects than γMnS due to the dissipation of energy by the lattice thermal vibration, which caused the surface-localized electrons to quench rapidly due to non-radiative defects, greatly reducing the fluorescence of semiconductor structures quantum yield, which resulted in weak NLO performance
Line 187 The authors say that “gamma-MnS was not as stable as that of alpha-MnS”. But I understood that the NLO of gamma MnS is better than alpha MnS. What does this sentence mean, or what ‘stable’ mean?
Response 9: This is our mistake. We remove that sentence and change as follow.
Nanopariticles are condensed and not scattered. Nanoparticles make up bulk crystals, so the massive crystals need to be dissolved in the solution with sufficient ultrasound to obtain nanoparticles.
In Fig. 6(a), for the a-MnS 10 and12, there seems many dips around Z=0. Does this has physical meaning or show merely noise?
Response 10: This has physical meaning and it is not noise. This is the NLO response which nanoparticles exhibited.
In Fig. 7, how the result of gamma-MnS compared with alpha MnS is expressed?
Response 11: The synthesis time of γMnS is the same as that of αMnS-6, but the synthesis temperature is different. Therefore, γMnS is compared with αMnS-6, while αMnS nanoparticles with different synthesis times are compared with themselves.
Minor points Line 126 “Figure 2(g) and (h)” -> “Figure 2 (d)” Line 132 “a-MnS-10 was 272 nm” -> “a-MnS-14 was 272 nm” Line 158 “OA Z-scan OA Z-scan” -> “OA Z-scan”
Response 11: We have corrected errors and similar errors in the manuscript.

Reviewer 2 Report
The authors have experimentally studied tunable nonlinear optical property of MnS nanoparticles with different size and crystal form. The paper is novel and clear. Although the paper is on MnS nanoparticles, the citation of previous literatures including the importance of other nanomaterials and comparison of their nonlinear coefficients is a must.
The reviewer thinks that the following points should be added, clarified or revised considering a revision:
- Line 105: Fig. 2 does not show what has been mentioned.
- Any description for Fig. 2d?
- Line 126: What are Fig. 2 (g) and (h)?
- OA and CA should be introduced.
- The parameters in the equations should be introduced (including their quantities).
- The figures (e.g., 6 & 7) must be well described within the text. The descriptions are now very concise.
- The ripples on z-scan curves should be justified.
- How much is the optical loss?
Author Response
Dear reviewer,
Thank you for your letter and for the comments concerning our manuscript entitled “Tunable nonlinear optical property of MnS nanoparticles with different size and crystal form” (ID: nanomaterials-666187). Those comments are all valuable and very helpful for revising and improving our paper, as well as the important guiding significance to our researches. We have carefully studied the comments and have made correction which we hope to meet with approval. Revised portions are marked in red in the manuscript. The main corrections in the manuscript and the responds to the reviewer’s comments are as follow.
Line 105: Fig. 2 does not show what has been mentioned.
Response 1: This is our mistake. And we corrected to “Fig. 1”.
Any description for Fig. 2d?
Response 2: The description for Fig. 2d is that “Green areas represented sulfur and red areas represents manganese, which indicated that product was MnS, not other substances”.
Line 126: What are Fig. 2 (g) and (h)?
Response 3: This is our mistake. We corrected in manuscript.
OA and CA should be introduced.
Response 4: We add them in manuscript.
Figure 5 (a) and (b) showed typical OA (open apture) and CA (close apture) /OA Z-scan curves for nanoparticles of different crystal forms.
The parameters in the equations should be introduced (including their quantities).
Response 5: We add the content in the manuscript.
T could be observed from figure 5 (a) and (b) that T=0.86 and T=0.83. And â–³Tp-v also could be obtained by figure 5 (a) and (b) that â–³Tp-v,αMnS-6=0.29 and â–³Tp-v,γMnS=0.3. β could be calculated by the equations that βαMnS-6=1.78×10-11mW-1 and βγMnS=2.14×10-11 mW-1 And n2 also could be obtained by equations that n2,αMnS-6=2.17×10-12 esu and n2,γMnS=2.48×10-12 esu. Reχ(3), Imχ(3) and χ(3) are displayed in figure 7.
The figures (e.g., 6 & 7) must be well described within the text. The descriptions are now very concise.
Response 6: We add more description in manuscript.
Value of T could be observed from figure 6 (a) and (b) that T=0.83, T=0.71, T=0.65, T=0.69 and T=0.84. And â–³Tp-v also could be obtained by figure 6 that â–³Tp-v,αMnS-6=0.29, â–³Tp-v,αMnS-8=0.42, â–³Tp-v,αMnS-10=0.5, â–³Tp-v,αMnS-12=0.81 and â–³Tp-v,αMnS-14=0.53. β could be calculated by the equations that βαMnS-6=1.78×10-11mW-1, βαMnS-8=2.17×10-11mW-1, βαMnS-10=3.73×10-11mW-1, βαMnS-12=4.52×10-11 mW-1 and βαMnS-14=3.98×10-11 mW-1. n2 also could be obtained by equations that n2,αMnS-6=2.17×10-12 esu, n2,αMnS-8=3.18×10-12 esu, n2,αMnS-10=3.84×10-12 esu, n2,αMnS-12=6.31×10-12 esu, n2,αMnS-14=4.08×10-12 esu. And Reχ(3), Imχ(3) and χ(3) are displayed in figure 7. The nonlinear absorption characteristics exhibited by the nanoparticles first increased and then decreased as the size of the nanoparticles decreased.
The nonlinear refraction characteristics of αMnS also decreased with the increase of size, which showed that changing size by controlling synthesis time could control nonlinear characteristics of nanoparticles. At the same time, it could be seen from figure 7 that changing the temperature to control crystal form of nanoparticles could also change nonlinear characteristics of nanoparticles. The nonlinear response of nanoparticles became controllable, which made them have application potential in the fabrication of nonlinear devices.
The ripples on z-scan curves should be justified.
Response 7: We changed as follow.
How much is the optical loss?
Response 5: A lot of measurements about nanoparticles showed that optical loss is less, and the light loss can be negligible. Light loss affects test results less.

Reviewer 3 Report
No comment. After English corrections the paper may be published
Author Response
Dear reviewer,
Thank you for your letter and for the comments concerning our manuscript entitled “Tunable nonlinear optical property of MnS nanoparticles with different size and crystal form” (ID: nanomaterials-666187). Thanks very much for the recognition of our article and useful suggestions.

Round 2
Reviewer 2 Report
The paper has now impoved much and reads better.
English language checks are required: eg. three order nonlinearilty , Rama? instead of Raman, deletion of some articles (the) and etc.